# The Effectiveness of Electrical Vestibular Stimulation (VeNS) on Symptoms of Anxiety: Study Protocol of a Randomized, Double-Blinded, Sham-Controlled Trial

**DOI:** 10.3390/ijerph20054218

**Published:** 2023-02-27

**Authors:** Teris Cheung, Joyce Yuen Ting Lam, Kwan Hin Fong, Yuen Shan Ho, Alex Ho, Calvin Pak-Wing Cheng, Julie Sittlington, Yu-Tao Xiang, Tim Man Ho Li

**Affiliations:** 1School of Nursing, The Hong Kong Polytechnic University, Hong Kong SAR, China; 2The Mental Health Research Centre, The Hong Kong Polytechnic University, Hong Kong SAR, China; 3Integrated Services for Persons with Disabilities, Christian Family Service Centre, Hong Kong SAR, China A; 4Department of Psychiatry, The University of Hong Kong, Hong Kong SAR, China; 5School of Biomedical Sciences, Ulster University, Coleraine BT52 1SA, UK; 6Department of Public Health and Medicinal Administration, Faculty of Health Sciences, University of Macau, Macau SAR, China; 7Department of Psychiatry, The Chinese University of Hong Kong, Hong Kong SAR, China

**Keywords:** vestibular stimulation, rct, anxiety, non-invasive brain stimulation, efficacy

## Abstract

The prevalence of symptoms of anxiety is increasing, especially during the COVID-19 pandemic. A home use transdermal neurostimulation device might help to minimize the severity of anxiety disorder. To the best of our knowledge, there is no clinical trial using transdermal neurostimulation to treat individuals with symptoms of anxiety in Asia. This gives us the impetus to execute the first study which aims at evaluating the efficacy of Electrical Vestibular Stimulation (VeNS) on anxiety in Hong Kong. This study proposes a two-armed, double-blinded, randomized, sham-controlled trial including the active VeNS and sham VeNS group. Both groups will be measured at baseline (T1), immediately after the intervention (T2), and at the 1-month (T3) and 3-month follow-up (T4). A total of 66 community-dwelling adults aged 18 to 60 with anxiety symptoms will be recruited in this study. All subjects will be computer randomised into either the active VeNS group or the sham VeNS group in a 1:1 ratio. All subjects in each group will receive twenty 30 min VeNS sessions during weekdays, which will be completed in a 4-week period. Baseline measurements and post-VeNS evaluation of the psychological outcomes (i.e., anxiety, insomnia, and quality of life) will also be conducted on all participants. The 1-month and 3-month follow-up period will be used to assess the long-term sustainability of the VeNS intervention. For statistical analysis, ANOVA with repeated measures will be used to analyze data. Missing data were managed with multiple mutations. The level of significance will be set to *p* < 0.05. Results of this study will be used to determine whether this VeNS device can be considered as a self-help technological device to reduce perceived anxiety in the general population in the community setting. This clinical Trial was registered with the Clinical Trial government, identifier: NCT04999709.

## 1. Introduction

Anxiety is a normal reaction to stress and is considered as an adaptive emotion to help prepare an individual towards perceived threatening situations or stimuli. Nevertheless, exaggerated anxiety without timely treatment may develop into anxiety disorder, which may have detrimental consequences affecting an individual’s quality of life, sleep quality [1], and daily functioning [2]. Before 2020, mental disorders were leading causes of the global health-related burden, with depressive and anxiety disorders being leading contributors to this burden. The COVID-19 pandemic has further increased the global prevalence of anxiety disorders. Nonetheless, there seems to be a lack of robust, novel and non-invasive interventions to reduce the negative detrimental impact caused by COVID-19 pandemic to the community-dwelling populations.

Global prevalence of anxiety disorders during the COVID-19 pandemic

Some researchers conducted a systematic review reporting the global prevalence of anxiety disorders during the COVID-19 pandemic between 1 January 2020 and 29 January 2021 in 204 countries. Two COVID-19 impact indicators, daily SARS-CoV-2 infection rates and reductions in human mobility, were associated with an increased prevalence of anxiety disorders (increase of 25.6% (23.2–28.0)). Females and younger ages were affected more by the pandemic than their male counterparts. It was estimated that anxiety disorders caused 44.5 million (30.2 to 62.5) DALYs globally in 2020 [3]. Recent findings derived from a meta-analytic review [4] also found that the global prevalence of anxiety was 29.57% (95% CI: 24.67–34.47).

Local research findings on anxiety before and during the COVID-19 pandemic

An older local cross-sectional study (Hong Kong Mental Morbidity Survey) (HKMMS) examined the prevalence of common mental disorders (CMD) in 5719 Chinese adults, aged between 16–75, in the general population before the COVID-19 pandemic. Results showed that the weighted prevalence estimate for any past-week CMD was 13.3%, with mixed anxiety and depressive disorder being the most frequent diagnoses. Gender being female, divorced/separated, alcohol misuse/substance dependence, lack of regular physical exercise, and a family history of mental disorder were significant correlates of CMD. Less than 30% of individuals with CMD consulted mental health services [5].

Another population-based local study examined the prevalence of anxiety and depression amidst the COVID-19 pandemic. Of the 500 random samples recruited, 19% had probable depression (PHQ-9 score ≥ 10) and 14% had anxiety (GAD score ≥ 10). In addition, 25.4% reported that their mental health had deteriorated since the pandemic. However, different psychological instruments were used in these studies with different sampling methods; therefore, results need to be interpreted with caution. Notwithstanding the limitations of these studies, it is evident that a significant proportion of Hong Kong citizens are at risk of anxiety during the pandemic era, either those in the medical profession or in the general population. Anxiety is closely linked with depression [6]. Anxiety may also increase sedentary behavior [7], influence arterial pressure, and reduce the activity of individuals’ immune system [8], all of which lead to negative impact on health. Untreated anxiety may spiral easily to depressive symptoms, further increasing the global disease burden and psychiatric morbidity.

The emergence of the coronavirus pandemic has led to the implementation of several precautionary measures across the globe. Hong Kong is of no exception. Hong Kong is one of the world’s densely populated regions, and this possibly explains why the Hong Kong SAR government has enacted stringent quarantine measures and large-scale lockdowns for major entertainment and sports amenities (e.g., amusement parks, public swimming pools, theme parks, etc.). Border closures, social distancing, proper hand hygiene, and compulsory face mask wearing in public, etc., have inevitably increased self-perceived anxiety levels during the pandemic era [9]. As of today, the COVID-19 pandemic in Hong Kong is still an ongoing public health concern, and a significant proportion of Hong Kong citizens are having pandemic fatigue, which increases the odds of COVID-19-related anxiety symptoms [10]. In Hong Kong, low socioeconomic districts were the epicenters of a third-wave outbreak of COVID-19 in July and August 2020, suggesting that people from low socioeconomic classes are vulnerable groups. Socially disadvantaged people are relatively more vulnerable to the physical, mental, and social impacts of infectious diseases. Notably, existing studies were largely cross-sectional, and participants’ narrative comments toward combating the COVID-19 pandemic are under-investigated. Nevertheless, a study [11] interviewed thirty-five economically disadvantaged individuals during the COVID-19 outbreak and examined how these individuals were further disadvantaged in the outbreak by delineating how health inequality intersected with social inequality. Semi-structured interviews were conducted from February to April 2020. Results showed that the soaring prices of disposable surgical face masks and other disinfecting products (hand sanitizer, alcohol preps, disinfectants) alongside other COVID-19-related social distancing policies imposed severe economic and psychological burdens on the participants. All these elements seemed to have interrelated effects which in turn limit accessibility to healthcare, leading to less positive health outcomes.

Existing empirical evidence of the Vestibular Neurostimulation (VeNS)

Since the 20th century, some promising research has demonstrated that VeNS can improve postural stability in patients with Parkinson’s disease [12,13,14], bilateral vestibulopathy [15], or patients of the elderly population [16,17]. Recent VeNS research also showed that repetitive stimulation of the vestibular system may influence hypothalamic control of metabolic homeostasis, and thus decrease fat storage. In other words, VeNS may well be considered as an alternate self-help treatment option for obesity, diabetes, or other related metabolic diseases [18]. Recent findings from meta-analysis also found that there was a positive association between diabetes and anxiety symptoms [19]. Such findings were also echoed by a recent metanalytic review that non-invasive brain stimulation had a significant effect on the reduction of anxiety symptoms [20]. Nonetheless, there seems to be no empirical evidence on the efficacy of VeNS on symptoms of anxiety in Asia, particularly in Hong Kong or China. This research gap has given us the impetus to execute this trial.

Vestibular Neurostimulation (VeNS)—Modius Stress (MS)

Modius Stress (MS) is a non-invasive, transdermal neurostimulation device with a battery-powered headset designed to transcutaneously deliver low-level electrical energy (0–1500 microamperes) to the subject’s head (neurostimulation) to treat anxiety. This MS delivers neurostimulation through two self-adhesive electrode pads which will be placed on the subject’s skin, overlying each mastoid process behind the ear. MS delivers a small electrical impulse which can be adjusted up or down by the subject using the buttons on the headset (max. 1.5 mA at 100 Hz) or via Bluetooth using the mobile app to a level (0–10) where the subject can feel the tingling sensation in the skin (Table 1).

The subject will determine the level of neurostimulation once they begin to experience the gentle swaying, indicating modulation of the vestibular nerve. This device will automatically turn off after 30 min of neuromodulation. The MS device will deliver a small electrical current onto the skin through the electrode pads. This electrical current serves as a neurosignaling waveform which modulates cranial nerves that influence the balance between the parasympathetic and sympathetic nervous systems. MS stimulates the head and vestibular nerves via Cranial Electrotherapy Stimulation (CES).

Cranial Electrotherapy Stimulation (CES) and past research using CES on anxiety

CES is a non-invasive and well-endured neuromodulation treatment that applies pulsed, alternating microcurrent transcutaneously to the head via electrodes placed on the subjects’ earlobes, mastoid processes, zygomatic arches, or the maxilla–occipital junction [21]. CES has been proven to be an effective treatment for anxiety in a study [22] that 83% of the participants (n = 150) had >50% reduction in the Hamilton Rating Scale for anxiety scores after 5 weeks of treatment and co-morbid depression. A recent study also showed that CES stimulation was highly effective on individuals with GAD, with 47.8% remission (n = 161), and a GAD-7 score < 7 after 24 weeks of CES stimulation [23].

A longitudinal follow-up study [24] assessed the effectiveness of vestibular stimulation in the management of stress in twelve female gastric ulcers patients who were given six months of vestibular stimulation. Results showed that these participants’ depression, anxiety, and stress levels were significantly decreased after 3 months of vestibular stimulation (*p* < 0.05) when compared with baseline scores, and these instrumental scores further decreased after 6 months of intervention (*p* < 0.05), indicating that vestibular stimulation is a long-term sustainable intervention in the management of gastric ulcers. Apart from VeNS and CES, Galvanic Vestibular Stimulation (GVS) is another non-invasive stimulation acting on the vestibular system.

Galvanic Vestibular Stimulation (GVS)

GVS is also a non-invasive method used to stimulate the vestibular system. The vestibular system includes the sensors, neural pathways, vestibular nuclei, and the cortical areas receiving integrated vestibular inputs. The key role of the vestibular system is in postural control or gaze stabilization and involves some cognitive functions and emotion processing. Nevertheless, GVS is known to induce motion sickness symptoms such as nausea. Several studies have revealed a modulating effect of vestibular stimulation on anxiety levels. For example, a study [25] evaluated the tolerability and efficacy of a GVS protocol on anxiety and tested the impact of stimulation parameters (duration) on anxiety. Twenty-two students underwent three stimulation conditions: (1) a sham session with no stimulation; (2) 38 min of GVS; and (3) 76 min of GVS. Results showed a significant diminution in anxiety levels for the second group (those receiving a 38-min session of GVS), while a low level of motion sickness was reported for the third group (those receiving a 76-min session of GVS). In this trial, we will focus on the efficacy of VeNS in the treatment of anxiety symptoms in the general population.

### 1.1. Objectives

The aims of this study are (1) to evaluate the efficacy of VeNS on anxiety severity on community-dwelling adults in Hong Kong and (2) to examine the association between VeNS data usage, anxiety symptom severity, insomnia severity, and quality of life.

### 1.2. Hypotheses

#### 1.2.1. Primary Hypothesis

Participants in the active VeNS group will have at least a 50% reduction in their (General Anxiety Disorder-7) (GAD-7) score after four weeks of treatment, and this will be maintained for the 12-week follow-up. However, participants in the sham VeNS group will have a <5% reduction in their GAD-7 score after the 4-week treatment and during the 12-week follow-up.

#### 1.2.2. Secondary Hypotheses

Participants who have higher data usage in the active VeNS group will have statistically significant changes in insomnia severity and improved quality of life after the 4-week treatment that will be maintained during the 12-week follow-up. There will be no change in insomnia and quality of life in the sham VeNS group immediately after treatment and at the post-treatment follow-up at 12-weeks.

## 2. Materials and Methods

### 2.1. Trial Design

This is a four-week VeNS treatment on anxiety among adults in the general population. The trial design complies with the Consolidated Standards of Reporting Trials (CONSORT) statement [26]. Participants will be randomized into the active VeNS or sham VeNS group. All the participants will be informed about the randomization procedures and that they have a 50% chance of receiving the VeNS or the sham VeNS. This study will be conducted in accordance with the Declaration of Helsinki [27]. Both groups will be measured at the baseline (T1), immediately after the 4-week intervention (T2), and during a 3-month follow-up (T4) [23,28] (see Figure 1).

### 2.2. Subjects and Sample Size Estimation

Since the collaborative UK study is ongoing, we thus have to take reference of a similar CES study on GAD [23] on ‘Clinical effectiveness and cost minimization model of Alpha-Stim cranial electrotherapy stimulation in treatment seeking pts with moderate to severe GAD’. Participants in the CES group had a significant reduction in GAD-7 score (from baseline 15.77 (3.21) to 10.14 (4.86) (Cohen’s *d* = 1.3)) after 4 weeks of CES treatment.

Considering this trial is a sham-controlled trial, we hypothesize a large effect in the MS group. We used G*power version 3.1.9.7 (https://stats.oarc.ucla.edu/other/gpower/) (accessed on 1 April 2022) nto calculate the target sample size. With a statistical power of 99% and a statistical significance level at 0.05 to detect a large between-groups effect size of 1.3, each group will require 33 subjects. With an estimated attrition rate of 30% during the 12-week post-treatment follow-up, a total sample of 66 is required in this trial.

### 2.3. Recruitment

Samples will be recruited from our collaborative universities and Christian Family Services Centre (CFSC) in Hong Kong. A QR code flyer will be flagged up in communal areas in the Hong Kong Polytechnic University (HKPU), University of Hong Kong (HKU), and the Chinese University of Hong Kong (CUHK). Email invitations with the QR code poster will also be sent to all staff/students and alumni across different faculties/departments at HKPU, HKU, and CUHK. The project poster will also be advertised on the Facebook and Twitter of the School of Nursing, HKPU.

### 2.4. Eligibility Criteria

To obtain a homogenous sample, the inclusion criteria will be: (1) having a GAD-7 score > 10; (2) ethic Chinese, aged 18–60 years; (3) able to understand/read Chinese; (4) having an Insomnia Severity Index < 14; (5) able to provide written informed consent; (6) having Wi-Fi and Bluetooth network in iOS/Android mobile phones; (7) able to attend the face-to-face demonstration session for proper use of the MS device and return-demonstrate to research assistants in the training venue (Integrative Health Clinic, School of Nursing, The Hong Kong Polytechnic University; (8) willing to engage with the project team on a weekly basis via zoom meetings/WhatsApp/telephone to ensure compliance, proper usage of the MS device and report any technical issues; (9) not undergoing any extreme lifestyle changes that may impact on sleep quality (e.g., dietary changes/increase/decrease exercise levels) throughout the study period; (10) not taking any prescribed/over the counter anxiolytics.

### 2.5. Exclusion Criteria

Individuals with: (1) history of eczema/skin breakdown/other dermatological condition (e.g., psoriasis) affecting the skins behind the ears; (2) HIV/AIDS infection, as HIV will lead to vestibular neuropathy; (3) use of beta-blockers/antidepressants/any other medications that may affect the neurostimulation; (4) history of stroke/epilepsy/severe head injury/neurosurgery; (5) active migraine with aura; (6) significant communicative impairments; (7) metal implant in brain or pacemaker, implanted defibrillator, deep brain stimulator, vagal nerve stimulator, etc.; (8) undertook corticosteroid treatment within the last six weeks before the first TPS treatment; (9) pregnancy or breastfeeding women; (10) cognitive impairment including Dementia/Alzheimer’s disease, and mild cognitive impairment; (11) history of Bipolar affective disorder, psychosis or substance use disorders; (12) regular use of antihistamine medication in the last 6 months (because histamine receptors are present in the vestibular system); (13) history of malignancy in the past 12 months; (14) a diagnosis of myelofibrosis/myelodysplastic syndrome; (15) history of vestibular dysfunction or inner ear infections/diseases; and (16) previous use of any VeNS device will be excluded in this study.

### 2.6. Screening and Self-Administered Questionnaire

Participants will complete an online application form soliciting sociodemographic information (age, gender, educational background, marital status, monthly household income, living circumstances, employment status, psychiatric history and duration of having anxiety symptoms (in years/months), currently taking prescribed or over the counter anxiolytics (yes/no), family history of anxiety disorder (yes/no), and other psychiatric disorders, etc.) before filling in the screening tools (GAD-7 and Insomnia Severity Index). Participants have to indicate their written consent in the online application form before proceeding to the screening procedures.

### 2.7. Randomization, Allocation, Blinding and Masking

All consenting participants will be listed in alphabetical order according to their surnames and each participant will be assigned a unique identifier. An independent statistician will use a computer-generated list of random numbers (www.random.org) (accessed on 1 April 2022) to ensure concealment of randomization. Randomization will be conducted by an independent statistician off-site using a stochastic minimization program to balance gender, age, and GAD-7 scores of the participants. Block randomization with blocks of 10 (total: 7 blocks) will be used to allocate treatment groups. Participants from each block will be randomly assigned to the active VeNS groups or the sham VeNS groups in a 1:1 ratio. To avoid information flow, participants and research associates will be blinded to the group allocation to minimize potential contamination of the effects of VeNS or subject bias. The Principal Investigator will not be involved in data collection or pre- and post-VeNS measurements. Outcome measurements will be conducted by the research associates. Participants will be asked to guess the grouping (active VeNS vs. sham VeNS) when filling in the face-to-face post-test survey after finishing the 4-week interventions to determine the success of subject blinding [29].

### 2.8. Interventions

In this study, all participants (in both the active and sham VeNS groups) will receive a VeNS device for home use in our Integrative Health Clinic, School of Nursing, HKPU, after receiving the device training by the research associates. Participants will receive 20 VeNS sessions in total in this study, with each session lasting for 30 min over a 4-week period (i.e., Monday–Friday, total treatment time: 10 h). We believe that a four-week VeNS intervention will be sufficient enough to test the efficacy of VeNS on symptoms of anxiety. Participants will be followed up immediately after post-stimulation during the 4-week and 12-week periods (Figure 1, CONSORT Flow Diagram). We believe that a post-stimulation follow-up of up to 3 months is sufficient to evaluate the sustainability of VeNS.

#### 2.8.1. VeNS Group

Subjects should open the study app (VESTAL) prior to each VeNS stimulation session. Subjects will put on the VeNS headset (see Figure 2) and apply two self-adhesive electrode pads on the mastoid process behind the ear. Subjects should remain in a sitting down and resting position throughout the stimulation period. Subjects will then turn on the device which will then deliver a small electrical impulse (range from 0 mA to max. 1.5 mA, level 0–10) at 100 Hz. Subjects can adjust the device output until they are aware of a comfortable tingling sensation in the skin. The VeNS will then deliver the stimulation for 30 min and will automatically turn off. The level of stimulation is determined by a gentle swaying indicating modulation of the vestibular nerve. Subjects will be able to check their activities on the study app and modify the stimulation level either via the mobile app or via the iPods. The device can be paused/stopped by pressing the power button twice on the headset or via the study app. Subjects will hear a single beep sound whenever they change the stimulation level. After the stimulation, subjects can remove the headset and dispose the electrode pads.

#### 2.8.2. Sham VeNS

Subjects in the sham-controlled group will follow the identical procedures as set in the VeNS group. Subjects will receive the initial stimulation for 30 s and then tap down to 0 mA for 20 s. Each session will deliver a sham VeNS stimulation at a frequency of 0.8 Hz to reduce the likelihood of a vestibular sympathetic response that may induce weight loss [30]. Subjects will also experience sensations of skin tingling and vestibular stimulation so that subjects in this group will believe that they have been allocated with an active device.

On completion of each VeNS session, subjects will charge the device using the micro-USB and charger provided by the Project team for usage in the next stimulation session. The device will automatically stop the stimulation after 30 min of usage per day and subjects will need to undergo a 16-hour lock-out period to reuse the device again.

Data will be collected via the Bluetooth of the device. Total usage per day, average intensity used, and average resistance will be logged and stored on an encrypted server. The study app will automatically upload the data usage to our backend server when connected to Wi-Fi. All individual data with a unique trial identifier will be allocated at the recruitment time and collected via REDCAP by the PI and recorded in the electronic CRF.

### 2.9. Adverse Events and Safety Checklist

Given the low current (max. 1.5 mA) and voltage (4.25 v) of the Modius Stress device, it is generally safe for home users. However, some subjects may have skin irritation at the electrodes’ sites, sensation of disequilibrium, nausea/vomiting, headache, and an electrical tangling sensation. Hence, an adverse events checklist will be used to monitor/quantify the occurrence of these events in this trial. Subjects are encouraged to contact the project team members should these adverse effects become serious which warrant medical attention. The PI will then assess the subjects’ physical and mental conditions and decide whether they could continue the intervention or not.

### 2.10. Fidelity

To ensure the fidelity of the intervention, the project team will ascertain whether the interventions will be delivered as designed. To ensure compliance, the Project team will monitor the duration of usage data on a weekly basis (Friday). If any subject’s weekly usage is <2.5 h, research associates will send a WhatsApp text message as a gentle reminder and investigate if the subject has experienced any technical issues in using the device. The PI will work closely with the research associates to optimize compliance throughout the study period.

### 2.11. Data Management

Participants’ data in both groups will be stored in two separate datasets with an identifier linking these data. Both sets of data will be encrypted using TrueCrypt (http://www.truecrypt.org) (accessed on 1 April 2022). The data from baseline, and the 4-week and 12-week follow-ups will be linked according to personal data. All precautions in data protection will be taken, as suggested by TrueCrypt. To prevent the leakage of personal data, only the PI will have access to the personal dataset. Written consent will be obtained from all participants. An information sheet containing the purpose of this trial and the VeNS procedures, as well as the Modius Stress leaflet, will be provided to all participants. Participants will be informed of their anonymity; withdrawal or non-compliance will not result in any consequences.

### 2.12. Outcome Evaluation (Primary and Secondary Outcomes)

The primary objective of this study is to evaluate the effects of VeNS on participants’ anxiety symptoms in Hong Kong. Secondary objectives include examining the effects between VeNS data usage on insomnia severity and quality of life. All the primary and secondary outcomes will be assessed at baseline and during the 4-week post-treatment and 3-month post-treatment periods for evaluation of immediate and short-term effects [27].

#### 2.12.1. Primary Outcome

##### Anxiety

Anxiety will be assessed using the General Anxiety Scale 7 (GAD-7) [31]. GAD-7 is a widely used, reliable self-reported measurement of generalized anxiety disorder. GAD-7 consists of 7 core items assessing nervousness; inability to stop worrying; excessive worry; restlessness; difficulty in relaxing; easy irritation and fear of awful things happening. Subjects are asked how often they have been bothered by each of the 7 core symptoms of generalized anxiety disorder in the last two weeks on a 4-point Likert scale from 0–3, with 0 = “not at all”, 1 = “several days”, 2 = “more than half the days”, and 3 = “nearly every day”. The total score is obtained by adding up the scores of all the 7 items. Score ranges from 0 to 21, with scores of ≥5, ≥10, and ≥15 representing mild, moderate, and severe anxiety symptom levels, respectively [28]. The GAD-7 has good reliability and validity in the Chinese population [32].

#### 2.12.2. Secondary Outcomes

##### Insomnia Severity

Insomnia will be assessed using the Chinese version of the Insomnia Severity Index (ISI) which consists of 7 items measuring the day and night symptoms of insomnia in individuals. This ISI consists of seven items (1: perceived difficulty; 2: falling asleep; 3: time of awakening; 4: satisfaction with current sleep pattern; 5: interference with daily functioning; 6: noticeability of others’ impact on lack of sleep; 7: degree of perceived distress/concern caused by the sleep problem). Subjects will rate each question on a 4-point Likert scale (0–3); score ranges from 0–28, with 15–21 indicating moderately severe insomnia. ISI has been used in the Chinese population with good psychometric properties, with Cronbach’s alpha 0·81 and item-to-total correlations in the range of 0.34–0.67 [33].

##### Quality of Life

Quality of life will be assessed using the Chinese version of the 36-item Short Form Health Survey (SF-36) [34]. SF-36 include 36 questions related to an individual’s QoL in eight scales: physical functioning (PF), role-physical (RP), bodily pain (BP), general health perceptions (GH), vitality (VT), social functioning (SF), role-emotional (RE), and mental health (MH). The raw scores for each scale are transformed to a scale ranging from 0–100, with higher scores indicating ‘better’ QoL. SF-36 is summarized in two component summary scores, the Physical Component Summary (PCS) and the Mental Component Summary (MCS). This Chinese version of SF-36 is a valid and reliable instrument, with the overall Cronbach’s α coefficient being 0.943, while the Cronbach’s α coefficients for each of the dimensions were all >0.70 [35].

## 3. Statistical Analyses

All statistical analyses will be performed using the statistical software R for Windows (R version 4.1.0). Means and standard deviations (SD) for the continuous variables will be presented, while numbers and percentages for the categorical variables will be shown. A *p*-value < 0.05 is considered statistically significant. Sociodemographic differences between the VeNS group and the sham VeNS group will be analyzed using the chi-square test and t-test. If there are significant differences between sociodemographic factors, covariates will be considered as confounding variables in the analyses. Normality of the primary outcome (GAD-7) scores will be tested using the Shapiro–Wilk test for each combination of factor levels (group and time). A t-test will be used to test the baseline difference. Analysis of variance (ANOVA) with repeated measures will be used to test the group (between-subject factor), time (within-subject factor), and group x time interaction effects of the GAD-7 score between the VeNS group and the sham VeNS group. Post hoc comparisons between groups and time points will be conducted using a *t*-test with the Bonferroni correction. Normality of the secondary outcome scores will be tested using the Shapiro–Wilk test for each time point. For normally distributed outcomes, ANOVA with repeated measures will be used to determine whether the outcome scores are significantly different between pre-and post-test periods. For outcome scores that are normally distributed, a non-parametric Friedman test will be used to test the mean difference. A Cohen’s d effect size for each outcome will be calculated, where d = 0.2, 0.5, and 0.8, corresponding to small, medium, and large effect sizes, respectively. Participants’ total usage time will also be used to investigate the effectiveness of the VeNS on anxiety and total usage time will be used as a continuous covariate in a linear regression model for GAD-7 total score. Missing data will be managed with multiple imputation [36] by adopting the last observation carried forward (LOCF) method.

## 4. Novelty and Significance of This Study

This is the first double-blind, randomized, sham-controlled trial evaluating the efficacy of the VeNS device on the reduction in anxiety symptoms in Hong Kong, and this is also a multi-site comparison study with the UK and India. Mental health promotion in primary care is pivotal to reduce the global prevalence of anxiety-related problems. If our findings emerging from this study can prove that the VeNS device can help reduce anxiety symptoms, it would bring about a major breakthrough in neuroscience research and create significant research and community impacts within the general population. Community-dwelling individuals can have a more user-friendly, novel but safe adjunct treatment option to ease their anxiety states, which in the long run, can improve their overall quality of life, biological and mental wellbeing. From stakeholders and health policymakers’ perspectives, improving mental health in the general populations will also help reduce the global disease burden.

## 5. Limitations and Challenges of This Study

Although previous CES studies have been proven as primarily brain-calming techniques, approved by the US Food and Administration for the treatment of anxiety, insomnia and depression [37], some home users however may have fears or technical phobia regarding the use of mild electrical currents to the human brain. Compliance and safety use also need to be closely monitored throughout the intervention period, especially if the sample is sourced from community-dwelling populations.

## 6. Conclusions

This is a collaborative study with the University of Ulster in the UK. Findings emerging from cross-cultural contexts may shed new insights into the efficacy of VeNS and add new knowledge in the treatment of anxiety in the East and West. More importantly, findings from this large-scale study can provide scientific evidence to affirm whether this VeNS device can be considered as a self-help technological device that reduces perceived anxiety in the community setting. If this study is successful, individuals may have another self-help treatment option to allay their anxiety at the primary care level and foster their mental health on a short-and long-term basis.

## Figures and Tables

**Figure 1 ijerph-20-04218-f001:**
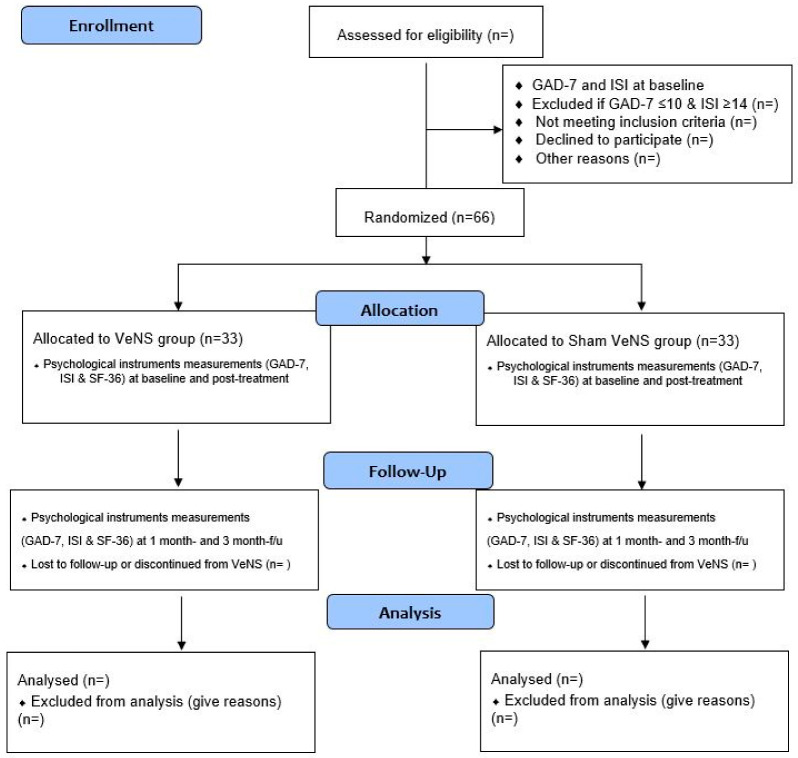
Flow diagram for participants’ enrollment, randomization, allocation, and follow-up.

**Figure 2 ijerph-20-04218-f002:**
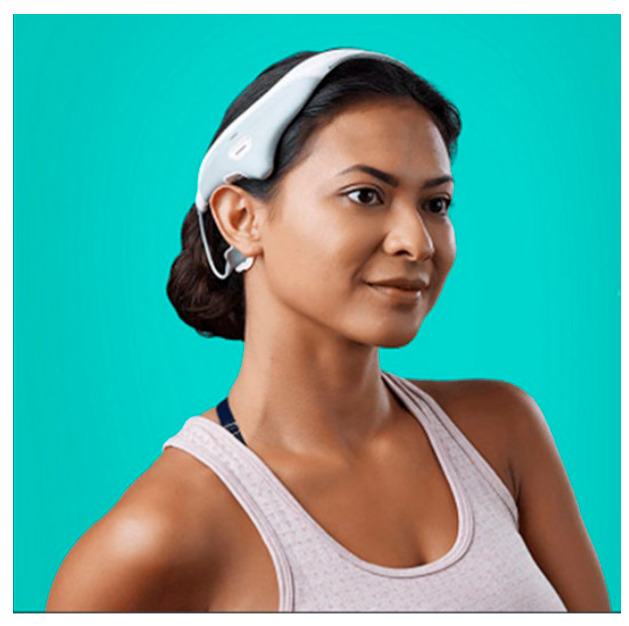
Modius Stress device (Image sourced from NEUROVALENS).

**Table 1 ijerph-20-04218-t001:** The electrical energy of each level.

Level	Stimulation Level
0	No stimulation applied (0 mA)
1–3	Stimulation applied (0.5 mA)
4–7	Stimulation applied (0.7 mA)
8–10	Stimulation applied (1 mA)

## Data Availability

The original contributions presented in the study are included in the article, further inquiries can be directed to the corresponding author.

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
