# Peer review of "The Effectiveness of Electrical Vestibular Stimulation (VeNS) on Symptoms of Anxiety: Study Protocol of a Randomized, Double-Blinded, Sham-Controlled Trial"

_ijerph, 2023, doi:10.3390/ijerph20054218_

Round 1
Reviewer 1 Report
Thank you for giving the opportunity to review this manuscript.
I think it is necessary to revise the manuscript before publication.
1) I do not think that previous studies on tDCS and rTMS were warranted. The study design of this meta-analysis was poor. These brain stimulations obtained no FDA approvals. I think it is better to delete the following sentences.
"Transcranial direct stimulation (tDCS) and repetitive transcranial magnetic stimulation (rTMS)
rTMS and tDCS are known effective noninvasive brain stimulation (NIBS) treatments for depressive symptoms. However, there is a paucity of research evaluating the efficacy of tDCS or rTMS on anxiety symptoms. A systematic review and meta-analysis [17] evauating the effectiveness of rTMS and tDCS in the treatment of anxiety disorders. Of the eleven papers retrieved (N=318), comprising participants assigned to tDCS or rTMS and to a sham-controlled group. Results showed a significant effect of NIBS in reducing anxiety scores in the active stimulation group compared to the sham-controlled group."
2) What is the biological mechanism of Electrical Vestibular Stimulation to treat anxiety disorders or to alleviate normal anxiety? Why Was this mechanism effective?
For example, some patients with depression presents hypoactivity of left prefrontal cortex, while high frequency rTMS over left DLPFC enhances the activity of the left prefrontal cortex, which contributes to the improvement in depressive symptoms. Furthermore, a single-session tDCS over DLPFC reduced amygdala threat reactivity ( Maria Ironside et al. Effect of Prefrontal Cortex Stimulation on Regulation of Amygdala Response to Threat in Individuals With Trait Anxiety: A Randomized Clinical Trial. JAMA Psychiatry. 2019 Jan 1;76(1):71-78.
Please explain the mechanism of Electrical Vestibular Stimulation to emphasize the rationale of this study.
3) I cannot understand the sentences of "CES has also obtained the FDA for the treatment of anxiety [21]."I cannot find any reference of [21], or any FDA approval of CES for the treatment of anxiety. Furthermore, A study of [19] just showed that CES improved anxiety with depression. I think it is still unclear that it is possible to improve anxiety only. In other words, no definitive treatment strategies have been established to manage anxiety, so novel treatment strategies including Electrical Vestibular Stimulation are warranted. Please explain the challenges of previous CES studies and emphasize the novelty and significance of this study.
4) Please change the title as "The Safety and Efficacy of electrical Vestibular Stimulation (VeNS) on Anxiety: Study Protocol for a randomized, double-blinded, sham-controlled trial." The safety of vestibular stimulation was not established. Normal anxiety is different from anxiety symptoms. Furthermore, Please change the sentence of "In this trial, we will focus on the efficacy of VeNS in treatment of anxiety symptoms in the general population." as "In this trial, we will focus on the safety and efficacy of VeNS to alleviate anxiety in the general population." In addition, Please change the sentences of "The aims of this study are (1) to evaluate the efficacy of VeNS on anxiety severity on community-dwelling adults in Hong Kong" as "The aims of this study are (1) to evaluate the safety and efficacy of VeNS on anxiety severity on community-dwelling adults in Hong Kong"
5) What does the following sentences mean?
"the device will deliver a small electrical impulse (range from 0mA to max. 1.5mA, level 303 0-10) at 100 Hz. Subjects can adjust the device output until they are aware of a comfortable tingling sensation in the skin."
Won't the investigators fix the intensity of the vestibular electrical stimulation in this protocol? Is it possible to reproduce this study? It is important to evaluate the treatment intensity to optimize the stimulation protocol.
6) Please attach the SIPRIT checklist and fill in the page numbers.
I think it is necessary to revise the manuscript.

Reviewer 2 Report
This manuscript describes a protocol for testing the effectiveness of electrical vestibular stimulation (VeNS) on anxiety symptoms, which is an ongoing clinical trial (NCT 04999709). The manuscript aptly describes VeNS and the promising results obtained from this technique in previous studies. The protocol in the current manuscript proposes to study the impact of VeNS on anxiety, insomnia, and quality of life using the General Anxiety Scale 7 (GAD 7), Chinese Insomnia Severity Index, and Chinese short-form health survey. This study aims to fill in the gap in data on the efficacy of VeNS in reducing Anxiety in the Asian population. Overall, this is a well-written manuscript, and addressing the below comments would further strengthen the manuscript.
Line 110: This is the first time VeNS is being introduced to the body of the introduction and hence it is necessary to explain what it stands for.
Line 110-180: Introduction: The structure of the introduction presents VeNS, tDCS, rTMS, CES, and GVS as standalone non-invasive brain stimulation (NIBS) techniques. It would also be better to have a comparative overview of the pros and cons in a comparative manner. This would help the reader understand why VeNS is an apt therapy to mitigate anxiety.
The study by Vergallito 2021 could be explained better to add to the rationale. Finally, the section could end with a description of the device that would be used in the study.
Line 169-183: Similar to the above comment, this section does not mention that VeNS uses GVS as the technology for stimulation. Kindly mention/explain this in the paragraph.
Line 272, section 2.7: The paragraph mentions that all consenting participants will be listed and randomized. Will there be no filtering based on sociodemographic data? Will the initial data obtained from participants be analyzed for eligibility? If yes, please mention the details in the section.
Line 293 and 297: The rationale for using a 4-week VeNS intervention and a 3-month post-stimulation follow-up is not clear. Kindly mention and support with reference why the authors believe that these time points should be studied.
Line 336: What do the authors mean when they say that adverse events are too unmanageable? What would be the threshold for the participants to report an adverse event? What would be the severity of the events that would be registered as adverse events? Clear definitions of severity and interventions in case of a severe adverse event should be reported.
LIne 397-398: What do the authors anticipate the immediate and short-term effects of the treatment to be? Kindly briefly mention this in the manuscript.
Reviewer 3 Report
Abstract
> The use of the future sense is particular. It should be repeated in this abstract that the paper concern a study protocol.
> The population of anxiety --> "The population of anxious adults"
> I don't understand the reference to the COVID-19 period in the abstract.
Keywords
> Are the used keywords the most relevant?
Introduction
At the end of the first paragraph, I suggest to the authors to add a transition that explains that their work focuses on the COVID-19 period (see the underlined sentences).
Concerning the paragraph entitled Global prevalence of Anxiety Disorders during the COVID-19 pandemic, I think that other important variables should be mentioned. The authors only mentioned 1 reference explain the prevalence of anxiety, there are much more which can enrich this point.
Lines 80 to 82 : I do not understand why the authors mentioned technology-based interventions. Sure, innovative technologies are highly relevant in psychology but the transition is quite abrupt. I suggest that the authors further develop the clinical thinking underlying that focus on technology-based interventions.
Lines 114 to 117: VeNS is interesting in physiological issues such as obesity, diabetes,...Ok, I understand. But I really do not understand the transition from this relevance to its use in anxiety. Can you further develop your hypotheses? This is quite unclear.
Can you add a figure of your device? This might help understanding how it works, where we should wear it,...
Objectives
> I am not sure to understand which kind of VeNS was used in this study? Several are described and the objectives mentioned that their effects are assessed on anxiety but this is not really clear.
> Why do you mention quality of life and sleep as outcomes? There have not been quite discussed in the beginning of your introduction. Can you further explain why these two variables are considered as important outcomes (more than depression, worries,...) ?
Materials and Methods
The description of the table 4 begins weirdly. Can you check the phrasing?
Lines 213 to 216: I don't understand.
Line 217: Participants in the CES group --> You mean the VeNS group? In the chart, you describe the groupe according to VeNS. It might be easier for your readers if you keep the same words.
Have you register your trial for RCT?
Overall, I think that the study protocol is interesting. I am not sure that the COVID-19 anxiety is relevant to be mentioned in this paper.
The manuscript should be checked for writing mistakes.

Reviewer 4 Report
The population of anxiety is increasing, especially in the COVID-19 situation. A home use transdermal neurostimulation device might help to minimize the severity of anxiety disorder. This study proposes a two-armed, double-blinded, randomized, sham-controlled trial including the active VeNS and sham VeNS group. All subjects in each group will receive twenty 30-min VeNS sessions during weekdays, which will be completed in 4-week period. Baseline measurements and post-VeNS evaluation of the psychological outcomes (i.e., anxiety, insomnia, and quality of life) will also be conducted on all participants. The 1-month and 3-month follow-up period will be used to assess the long-term sustainability of the VeNS intervention. For statistical analysis, ANOVA with repeated measures will be used to analyse data. Missing data were managed by multiple mutations. Results of this study will be used to determine whether this VeNS device can be considered as a self-help technological device to reduce perceived anxiety in the general population in the community setting. This manuscript was quite well written, but there were some problems that need to be polished drastically.
Minor comment:
Introduction
Transcranial direct stimulation (tDCS) and repetitive transcranial magnetic stimulation 120 (rTMS)
Q1.This section is not related to this manuscript, it is recommended to delete.
Galvanic Vestibular Stimulation (GVS)
Q2. This section is not related to this manuscript, it is recommended to delete.
Materials and Methods
2.1. Trial Design
Table 4. week VeNS treatment on anxiety among adults in the general population.
Q3.It must be a typo and needs to be corrected.
2.8. Interventions
We believe that a four-week VeNS intervention will be sufficient enough to test the efficacy of VeNS on symptoms of anxiety.
Q4. What is the author's basis for this statement?
Author Response
Response to Q1:
We have deleted the following paragraph from the main text.
“Transcranial direct stimulation (tDCS) and repetitive transcranial magnetic stimulation (rTMS)
rTMS and tDCS are known effective noninvasive brain stimulation (NIBS) treatments for depressive symptoms. However, there is a paucity of research evaluating the efficacy of tDCS or rTMS on anxiety symptoms. A systematic review and meta-analysis [17] evaluating the effectiveness of rTMS and tDCS in the treatment of anxiety disorders. Of the eleven papers retrieved (N=318), comprising 154 participants assigned to tDCS or rTMS and 164 to a sham-controlled group. Results showed a significant effect of NIBS in reducing anxiety scores in the active stimulation group compared to the sham-controlled group.”
Response to Q2:
Thank you for your recommendation but one of the reviewers suggested us to have a comparative overview of different types of neurostimulation in this article and GVS explains the mechanisms of VeNS in this study.
Response to Q3:
It was a slip, and we have reverted to “this is a four-week VeNS treatment on anxiety among adults in the general population.
Response to Q4:
Existing research have used 5 mins to 20 mins neuromodulation device in each intervention section on different healthy and disease populations (Utz et al., 2010), and in this study, we proposed a four-week VeNS stimulation (10 treatment hours) to determine whether this VeNS device can lead to a 50% reduction of the GAD-7 score.
Ref: Utz, K. S., Dimova, V., Oppenländer, K., & Kerkhoff, G. (2010). Electrified minds: Transcranial direct current stimulation (tDCS) and Galvanic Vestibular Stimulation (GVS) as methods of non-invasive brain stimulation in neuropsychology—A review of current data and future implications. Neuropsychologia, 48(10), 2789–2810. https://doi.org/10.1016/j.neuropsychologia.2010.06.002
Round 2
Reviewer 1 Report
Thank you for revising the manuscript.
I think this manuscript would be suitable for publication.